# Mixed Matrix Membranes Containing a Biphenyl-Based Knitting Aryl Polymer and Gas Separation Performance

**DOI:** 10.3390/membranes11120914

**Published:** 2021-11-23

**Authors:** Raquel Martinez-Tirado, Nastasia Yuriychuk, Marta Iglesias, Mar López-González, Eva M. Maya

**Affiliations:** 1Departamento de Química-Física de Polímeros, Instituto de Ciencia y Tecnología de Polímeros (ICTP-CSIC), Consejo Superior de Investigaciones Científicas, c/Juan de la Cierva 3, 28006 Madrid, Spain; raquel.mtez.tdo@hotmail.com (R.M.-T.); nasta@ictp.csic.es (N.Y.); 2Departamento de Nuevas Arquitecturas en Química de Materiales, Instituto de Ciencia de Materiales de Madrid (ICMM), Consejo Superior de Investigaciones Científicas (CSIC), c/Sor Juana Inés de la Cruz 3, Cantoblanco, 28049 Madrid, Spain; marta.iglesias@icmm.csic.es

**Keywords:** knitting aryl polymer, polysulfone, Matrimid, polyphenylene oxide, mixed matrix membrane, gas separation

## Abstract

Novel mixed matrix membranes (MMMs) were prepared using Matrimid (M), polysulfone (PSF) or polyphenylene oxide (PPO) as the continuous phase and a porous biphenyl-based knitting aryl polymer as a filler, synthesized through the Friedel–Craft reaction. The filler had little influence on the thermal and morphological properties of the membranes but affected the mechanical and gas transport properties, which were different depending on the type of matrix. Thus, in the case of MMMs based on Matrimid, the filler increased considerably the permeability to all gases, although no improvements in selectivity were achieved. A PSF-based MMM showed minor permeability increases, but not in all gases, while the selectivity was particularly improved for hydrogen separations. A PPO-based MMM did not exhibit variation in permeability nor in permselectivity with the addition of the filler.

## 1. Introduction

The presence of fillers based on porous organic polymers (POPs) in gas separation membranes yield an interesting class of mixed matrix membranes because the organic composition of the filler offers better compatibility and adhesion with organic polymeric matrices. Thus, the dispersion of these fillers in the matrices are much better than using inorganic fillers; these MMMs show higher gas permeability coefficients than the neat polymeric matrices, while the selectivity was maintained or increased [1,2]. Furthermore, in some cases, the presence of POPs has served to slow down the ageing of the membranes [3,4,5].

Polysulfone and Matrimid (Figure 1) are two of the most widely used polymers in gas separation technology and mixed matrix membranes containing inorganic fillers, with quite a number of reviews having been published about them [6,7,8,9]. The incorporation of porous organic polymer (POP) fillers into these matrices has also been explored: for example, porous polyimides fillers and benzimidazole-linked polymers (BILPs) have been used as fillers in Matrimid matrices [1,10]; hollow ionic covalent organic polymers and perylene-based polymers in polysulfone matrices [11,12]; and covalent triazine frameworks in both [13]. Another very interesting type of matrix with which to prepare mixed matrix membranes for gas separation is polyphenylene oxide (PPO) (Figure 1), a polar and semi-crystalline polymer. It is generally accepted in membrane technology that permeability decreases when the degree of crystallinity of polymers is increased, and that the transport of gas molecules occurs only through the amorphous phase [14,15]. However, the addition of inorganic fillers into semi-crystalline PPO matrices can produce amorphous PPO and therefore improve the gas transport properties [16]. 

In 2011, a new class of porous organic polymers (POPs) emerged onto the scientific scene: knitting aryl polymers (KAPs) [17], prepared from aromatic monomers without previous functionalization, which is a great synthetic benefit compared to conventional POPs that need tetra- or tri-functionalized monomers [18]. The first KAPs incorporated into gas separation membranes were reported in 2018 and were based on tryptycene units [19]. Those amino-functionalized tryptycene-based KAPs showed an increase in CO_2_ permeability up to 2.23 times that of the neat matrix, a polymer of intrinsic porosity with good gas separation performance, PIM-1 [19]. The next year, KAPs containing chloro-p-xylene units and their amino-functionalized derivatives were introduced into polyimide and polysulfone matrices [20]. In this case, the membranes containing un-functionalized KAP exhibited better and higher CO_2_ permeability coefficients than the corresponding membranes filled with an amino-functionalized derivative, although the latter were much more selective.

We have recently reported the use of biphenyl-based KAPs (K2Ph) and functionalized biphenyl-based KAPs with nitro and amino groups as fillers of a polycarbonate matrix [21]. In our study, the membrane containing the un-functionalized filler also showed the highest gas permeation coefficients of the series, while the selectivity was maintained. Based on these results, we decided to go one step further and to explore the effect of this non-functionalized KAP (K2Ph) (Figure 1) in other matrices, specifically in Matrimid (M), polysulphone (PSF) and polyphenylene oxide (PPO) (Figure 1), selected for the reasons mentioned above.

## 2. Materials and Methods

### 2.1. Materials

Matrimid 5218 (M, Figure 1) with a density of 1.2 g/cc was supplied by Ciba-Geigy (Basel, Switzerland). 

Polysulphone (PSF, Figure 1) in pellet form with a density of 1.24 g/cc was provided by Sigma-Aldrich (Steinheim, Germany). 

Polyphenylene oxide (PPO, Figure 1) with a density of 1.06 g/cc was provided by Poly Sciences (Warrington, PA, USA) 

The solvents used for MMM preparation were of high quality (>99%).

Oxygen and hydrogen (99.999% purity), nitrogen (99.9999%), carbon dioxide (99.998%), methane (99.9995%) and ethylene (99.995%) supplied by Praxair were used in permeation experiments.

### 2.2. Biphenyl-Based Knitting Aryl Polymer Filler (K2Ph)

A solution of 0.200 g (1.29 mmol) of biphenyl in 30 mL of dichloroethane was placed in a Schlenk flask and purged with nitrogen for over 20 min. Then, 2.09 (12.9 mmol) of FeCl_3_ was carefully added, followed by 2.28 mL (25.8 mmol) of formaldehyde dimethyl acetal (FDA). The mixture was stirred for 72 h at 80 °C. The resulting dark-brown precipitate was collected by filtration and washed three times using the following protocol: stirred with ammonia for 24 h to remove unreacted FeCl_3_; filtration; stirred with diluted HCl (4 mL of concentrated HCl in 100 mL of water) for 24 h for neutralization; filtration; stirred with hot methanol for 24 h; filtration; stirred with THF for 2 h; filtration; and stirred with acetone for 2 h. Finally, a brown product was obtained as fine powder and was dried at 120 °C overnight. Yield: 80%.

### 2.3. Mixed Matrix Membrane Preparation

Of the corresponding matrix (M, PSF or PPO), 400 mg was dissolved in 4 mL of chloroform. Then, 80, 40 or 20 mg (20, 10 and 5% weight) of the polymeric fillers were dispersed in the above solutions by controlled addition. After each small addition, the dispersion was stirred in an ultrasonic bath for 10 min, and afterwards underwent magnetic stirring for another 20 min. This procedure is repeated until the total addition of the filler (around 6–8 h); finally, the dispersion was stirred overnight under magnetic stirring.

The MMMs were prepared differently, depending on the matrix. For M- and PSF-based membranes, the corresponding dispersions were poured on a levelled silylated glass plate, allowing the solvent to evaporate at room temperature overnight. For PPO- based membranes, the corresponding dispersion was placed on a cellophane surface at 50 °C, allowing the solvent to evaporate at this temperature for 4 h, after which the film was removed. In all cases, a confinement ring was used to control the film’s thickness, which was between 90 and 120 µm. Then, M-, PSF- and PPO-based MMMs were dried in a vacuum oven overnight at 230, 190 and 180 °C, respectively.

The MMMs were designed with the name of the matrix, followed by the name and amount of the filler: M@K2PhX%, PSF@K2PhX% and PPO@K2PhX%. Pristine M, PSF and PPO membranes were prepared as described for the mixed matrix membranes, but in the absence of filler. The membranes obtained after solvent evaporation were dried in the above conditions.

### 2.4. Characterization Techniques 

Scanning electron microscopy (SEM) micrographs were acquired with a Hitachi SU-8000 microscope using a field emission filament at 0.5 kV. The membranes were fractured under liquid nitrogen and the cross-section was observed with a magnification that varied from 400 to 60 K.

Thermogravimetric (TGA) analyses were done in a TQ-500 apparatus from TA Instruments under an air atmosphere at a heating rate of 10 K min^−1^, from 50 to 900 °C.

FTIR-ATR spectra were recorded with a PerkinElmer spectrometer fixed to a device of attenuated total reflectance (ATR, Waltham, MA, USA). Spectra were recorded with a resolution of 4 cm^−1^ in a spectral range of 600–4000 cm^−1^.

Differential scanning calorimetry (DSC) was performed in a PerkinElmer Pyris 1 at 20 °C/min and in an inert atmosphere. The membranes were encapsulated in regular aluminum DSC pans and were heated from the ambient temperature to 300 °C (for PSF- and PPO-based membranes) and 350 °C (for M-based membranes) at a scanning rate of 20 °C/min in the first cycle to obtain the thermal properties of the membranes. Then, the membranes were cooled at the same rate to 25 °C, and the glass transition temperatures, Tg, were obtained in the second heating cycle from the inflexion of the heat flow versus temperature curves.

X-ray diffractograms (XRDs) were recorded with a BRUKER D8 Advance, with a radiation of CuK α = 1.5415 Å in the range of diffraction angles from 1° to 80°. The *d*-spacing was calculated by Bragg’s law according to the following equation:(1)nλ=2d sinθ
where *n* is an integer, *λ* is the wavelength of the X-ray radiation used, *d* is the *d*-spacing and *θ* is the measured diffraction angle.

The porous properties were determined by nitrogen adsorption/desorption isotherms at 77 K using a Micromeritics ASAP 2020 Micropore Dry, the BET technique was used to calculate the specific surface area and the N2-DFT method was used to determine the pore size average. The samples were degassed for 12 h by heating them to 100 °C under a vacuum before analysis.

The particle size distribution was measured by dynamic light scattering (DLS) using a Malvern Zetasizer Nano ZS (Malvern Instrument Ltd., Malvern, UK), equipped with a He-Ne laser beam of 4 mW and *λ* = 633 nm. Malvern Dispersion Software was used for data acquisition and analysis, applying a general-purpose algorithm to calculate the size distribution. The purified fillers K2Ph-FC and K2Ph-S were dispersed in filtered deionized water using an ultrasound bath. Solutions of 1 mg/mL were measured at 20 °C, and the particle size was averaged over at least 6 runs.

The skeletal density of the MMMs was determined by a helium pycnometer in an AccuPyc 1330 at 25 °C, using around 200 mg of the membrane. The mean value of three measurements was taken for each membrane.

The bulk density, which considers both the solids and the pore space of the membranes, was determined in an analytical Sartorius hydrostatic balance by weighing the samples in the air (W_air_) and in a liquid with a well-known density (W_liquid_): isooctane for PSF- and M-based membranes and water for PPO-based membranes. The mean value of four measurements was taken for each membrane. According to Archimedes’ principle, the density of the samples was calculated from: *ρ*_sample_ = *ρ*_liquid_ [(W_air_ − W_liquid_)/W_air_](2)

The bulk density (bulk density) was calculated:(3)$$\frac{1}{\rho_{bulk}}=V_{poro}+\frac{1}{\rho_{esqueleto}}$$

### 2.5. Gas Transport and Separation Properties

Permeation experiments of oxygen, nitrogen, carbon dioxide, hydrogen, methane and ethylene were carried out in a laboratory-made permeator [22] that consisted of a stainless steel cell kept inside a water thermostat at the temperature of interest, 30 °C. The polymer membranes were put inside the two cell-separating compartments, the high-pressure or upstream chamber and the low-pressure or downstream chamber. A high vacuum (about 10^−6^ bar) was made in the two semicells with a Pfeiffer turbomolecular pump and maintained overnight in the permeation device to remove the last traces of solvent and gas in the membranes. After that, the gas at a given pressure was suddenly introduced from the upstream chamber, provided with a Geometrics pressure sensor (0–10 bar), to the downstream chamber, where the evolution of the gas pressure with time was monitored via a personal computer with an MKS Baratron Type 628B pressure sensor operating in the range of 10^−4^–10 mmHg. The air inlet into the low-pressure chamber was also monitored as a function of time before each experiment and subtracted from the curves representing the flux of the permeant through the membranes against time in the downstream chamber. After each experiment, the vacuum was also maintained to remove the traces of the measured gas. The diffusion area of the membranes was 3.464 cm^2^.

As usual, the variation in the pressure of gas versus time presents a transitory process followed by a straight line (t→∝) corresponding to the steady-state flow behavior. In such cases, the permeation measurements can be described by the integration of Fick’s second law using appropriate boundary conditions [23]:(4)$$p\left(t\right)=0.2876\;\frac{p_{0}AlST}{V}\left[\left(\frac{Dt}{l^{2}}-\frac{1}{6}\right)-\frac{2}{\pi^{2}}\overset{\infty }{\underset{n=1}{\sum }}\frac{{\left(-1\right)}^{n}}{n^{2}}\exp\left(-\frac{n^{2}\pi^{2}D}{l^{2}}\;t\right)\right]$$
where *A* (in cm^2^), *l* (in cm), *V* (in cm^3^) and *T* (in K) are, respectively, the area of permeation, the thickness of the membrane, the volume of the downstream chamber and the absolute temperature. The variables *p_0_* and *p* (in cmHg) are the pressure of the up- and downstream chamber, respectively.

The intersection of that straight line with the abscissa axis is the time lag (*θ*), which is related to the apparent gas diffusion coefficient, *D*, by the equation suggested by Barrer [24]:(5)$$D=\frac{l^{2}}{6\theta }$$

The permeation coefficient, *P*, can be obtained from the slope of that straight line by means of the following expression:(6)$$P=\frac{273}{76}\frac{Vl}{p_{0}TA}\;\underset{t\to \infty }{\lim}\frac{dp}{dt}$$

Additionally, it is usually given in barrer [1 Barrer = 10^−10^ cm^3^ (STP) cm cm^−2^ s^−1^ (cmHg)^−1^].

Finally, the apparent solubility coefficient, *S*, can be calculated from *P* and *D* with the following expression:(7)S=P/D

Transport coefficients *D* and *S* are usually given in cm^2^, s^−1^ and cm^3^ (STP) cm^−3^ (cm Hg)^−1^, respectively.

The ideal selectivity of a gas, *A*, with respect to another, *B*, can be expressed as:(8)$$\alpha \left(\frac{A}{B}\right)=\frac{P\left(A\right)}{P\left(B\right)}=\frac{D\left(A\right)}{D\left(B\right)}\;\frac{S\left(A\right)}{S\left(B\right)}$$
where *P(A)* and *P(B)* are the permeability coefficients of gases *A* and *B*, respectively. In all cases, *P(A) > P(B).*

The absolute errors, *ε*, involved in the determination of *D* by the time lag method were obtained by means of the following expression:(9)$$\varepsilon =\left(\frac{\left|L\;\epsilon \left(L\right)\right|}{3\;\theta }+\frac{\left|\epsilon \left(\theta \right)\right|\;L^{2}}{6\;\theta^{2}}\right)$$

Additionally, the relative errors can be expressed as Δ = *ε*/*D*.

In the case of the mixed matrix membranes, two films were prepared in the same conditions described previously, and gas permeation tests were performed. The gas transport experiments were repeated two or three times depending on the membrane nature and gas.

The error involved in the determination of the diffusion coefficient lied in the range of 2 to 12%, depending on the nature of the polymer base (around 4–5% for PSF-; 2–3% for PPO-; and 11–12% for M-based membranes) and the percentage of load added in each MMM. Standard deviations lower than ±0.2 were found for the permeability coefficients.

The nature of the gas is another factor to take into consideration, particularly for the most condensable one, such as ethylene, which also has the greatest kinetic diameter and the lowest diffusion coefficient. In this case, it was necessary to confirm that a steady-state condition was reached. To do that, the values of *P* and *D* obtained by fitting Equation (4) were compared with the data obtained from Equations (6) and (5). For example, in an M@K2Ph-20% membrane, the values for permeability and diffusion coefficients of ethylene calculated from these previous equations were 0.73 barrer and 2.8 × 10^−10^ cm^2^/s, respectively. These values fit very well with those obtained by fitting Equation (4) to experimental results, 0.79 barrer and 2.3 × 10^−10^ cm^2^/s.

## 3. Results and Discussion 

### 3.1. Biphenyl-Based Knitting Aryl Polymer Filler (K2Ph)

Two different strategies have been reported for the preparation of knitting aryl polymers: one based on a Friedel–Craft reaction using formaldehyde dimethyl acetal (FDA) as an external cross-linker, FeCl_3_ as the catalyst and an 80 °C reaction temperature [17], and the other based on the Scholl reaction, in which a chlorinate solvent acts as both a solvent and cross-linker simultaneously, in the presence of AlCl_3_ as the catalyst and a 40 °C reaction temperature [25]. We previously prepared the biphenyl-based knitting aryl polymer filler (K2Ph) using the latter procedure [26]. To explore the effect of changing conditions, for this work K2Ph was prepared using the Friedel–Craft reaction. The first difference between the two procedures has been the purification of the resulting polymer. The K2Ph prepared by the Scholl reaction (using AlCl_3_ as the catalyst) was easily purified with diluted HCl over 60 min to remove the aluminum residues, and then was washed sequentially by being stirred with water, methanol, THF and acetone [21,26]. However, K2Ph prepared by the Friedel–Craft reaction (using FDA as a cross-linker and FeCl_3_ as the catalyst) was very difficult to purify. Initially, the thermogram of K2Ph recorded in an air atmosphere (Figure 2a) indicates 37% of residue, attributed to the formation of iron oxide, which indicated the presence of a large amount of iron in the polymer. A preliminary purification cycle was applied that involved the following sequences of filtering and washing: stirred with ammonia for 24 h to remove unreacted FeCl_3_; stirred with diluted HCl for 24 h for neutralization; stirred with hot methanol for 24 h; stirred with THF for 2 h; filtration; and stirred with acetone for 2 h. After this intense purification, it was only possible to reduce the residue to 13% (Figure 2b). Thus, two more purification cycles were necessary to almost completely eliminate iron residues (Figure 2c). The thermal stability of 2KPh was slightly lower (300 °C) than that of the sample prepared by Scholl coupling (330 °C) [26].

The structural characterization data of the filler prepared by the Friedel–Craft reaction were practically coincident with those reported for the filler prepared by the Scholl reaction [26]. However, the porosity properties were different. The N_2_ adsorption/desorption isotherms (Figure 3a, Table 1) revealed that the new K2Ph filler, prepared by the Friedel–Craft reaction (K2Ph-FC), has a lower BET surface area (1034 m^2^/g) than that produced by the Scholl reaction (1481 cm^2^/g), although the isotherm has less hysteresis, which would indicate greater regularity in the size and shape of the pores. The pore distributions, calculated by the N2-DFT method (Figure 3b), of K2Ph-FC mainly show a fraction of mesopores centered at 3.87 nm, whereas K2Ph-S (prepared by the Scholl method) contains mesopores with a size range of 2–4 nm and a smaller fraction of micropores centred around 1.5 nm. The porous properties and density values of both fillers are shown in Table 1. As might be expected, the bulk density estimated by Equation (3) is lower than the skeletal density determined by a He pycnometer. In both cases, the pores of this polymeric filler occupy 53% of its total volume.

The size of both fillers, K2Ph-S and K2Ph-FC, were measured by dynamic light scattering (DLS) in deionized water. As can be seen in Figure 4, the analysis revealed a unimodal Gaussian distribution for K2Ph-S with an average particle size of 0.11 μm, although the sample was too aggregated for DLS measurements. However, the K2Ph-FC filler showed good dispersion and bimodal Gaussian behavior, and an average particle size of 0.28 and 1.10 μm, respectively, was observed. These sizes are small enough to ensure a good dispersion of the load in the matrices, as has been previously observed in other MMMs [1].

The X-ray diffractogram of purified and dry K2Ph-FC (Figure 5a) shows some peaks of crystallinity at 4.9, 3.3, 3.0 and 2.5 Å, which indicates a certain order in this porous polymer. The peak at a lower angle (4.9 Å) can be attributed to the chain-to-chain distance of more-efficient polymer chain spacing responsible for the micropores, whereas the peak at 3.3 Å can be assigned to the periodicity expected from aromatic systems [27,28]. Thus, it could be presumed that the Friedel–Craft reaction led to a greater ordering of the benzene rings, decreasing the specific surface area and the pore volume compared to that synthesized by the Scholl reaction.

### 3.2. Membrane Preparation and Characterization 

As has been widely reported, the main difficulty in preparing mixed matrix membranes (MMMs) is reaching a good dispersion of the loads in the polymer matrices because of the great tendency of the fillers to agglomerate [29,30,31]. In this work, we employed the procedure that we recently reported [21], which consists of the controlled addition of K2Ph using small amounts of filler that are added to the solution of the polymeric matrices. After each addition, the resulting dispersion is stirred for 10 min with ultrasound and 20 min with magnetic stirring. The complete addition of the filler was made in a period of 6–8 h. Once the dispersions were ready, the membranes were prepared by casting. The M- and PSF-based membranes were obtained at room temperature, allowing for slow solvent evaporation overnight. However, the morphology of PPO films depends a lot on the solvent and evaporation temperature, which also affects their performance [32]. In our case, if the evaporation of the solvent is done at room temperature the PPO crystallizes and the membrane becomes opaque as well as brittle, and breaks. By heating it to 50 °C, the solvent evaporated more quickly, avoiding the crystallization of PPO and the formation of bubbles. 

Depending on the dispersion of K2Ph in the polymeric solution as well as its affinity with the polymer, each matrix admitted a different percentage of filler. Initially, M was loaded with 10% and 20% of filler, to determine the effect of the amount of that filler in the transport properties of MMMs, yielding the membranes M@K2Ph-10% and M@K2Ph-20%, respectively. According to the gas transport properties results, PFS was directly loaded with 20% of filler, yielding such a membrane as PSF@K2Ph-20%. However, PPO could not be loaded with either 20 or 10% of filler, allowing it to accommodate only 5%, resulting in the PPO@K2Ph-5% membrane.

Through X-ray diffraction measurements (Figure 5), the membrane preparation methods were validated. As can be observed, all membranes exhibited an intense amorphous halo, which indicated that the filler did not modify the amorphous nature of pristine polymers, PSF and M, in the mixed matrix membranes. In the case of PPO, Khulbe et al. [33] described that the PPO crystallization changed according to the solvent used and the technique employed for removing it. In our case, as can be observed in Figure 5d, the diffractogram of PPO film showed a broad amorphous peak and a shoulder at about 2θ = 14° and 23°, which correspond to a d-spacing of 6.3 and 3.9 Å, respectively, and could be assigned to intra- and intermolecular distances between the polymer chains, respectively [34]. This behavior is typical of PPO films prepared using chloroform as the solvent. The diffraction scan for the PPO-based mixed matrix membrane indicated that the mentioned casting protocols used in this case keep films amorphous, which is necessary to transport gases as crystallinity worsens gas permeability [35]. This result is especially interesting in this case since it is the only semi-crystalline polymer of the series. Besides, much less intense peaks at 2θ = 30° and 36° appeared in the diffractograms of the PPO@K2Ph-5% membrane, which could be associated with the appearance of a certain order in the filler, as has been mentioned previously. 

Once the MMMs were prepared they were characterized to determine the effect of the load in each of the matrices. The SEM images of the cross-section of the membranes (Figure 6) showed good compatibility between the matrices and the corresponding filler percentages. A homogeneous dispersion of K2Ph through the cross-sections of the membranes can be observed, and no defects or voids can be appreciated between the matrices and the filler. 

The thermal stability of the MMMs was examined by thermogravimetry (Figure 7 and Table 2). As can be observed, the filler K2Ph slightly decreases the thermal stability of all matrices. The M- and PPO-based membranes (Figure 7a,c) have the same degradation behavior with or without a filler, with one-step or two-step decomposition patterns, respectively. However, the pristine PFS matrix has a well-differentiated two-step decomposition pattern that is approximated by the presence of the load (Figure 7b).

Glass transition temperatures (Tgs) were determined by DSC (Figure 8 and Table 2). The Tgs of the pristine membranes are 190 °C (PSF), 326 °C (M) and 216 °C (PPO), values similar to those reported in the literature [36,37,38]. The K2Ph filler had a similar effect on all matrices, in the sense that the Tg of the mixed matrix membranes increased by 2–4 °C with the presence of the filler in comparison with the pristine polymers. Although these changes in Tg are not large enough and in some cases are within the experimental error, they might indicate a certain rigidity in the membranes that could be attributed to the introduction of the polymer chains inside the pores of the load, restricting its movement, as has been observed previously in MMMs of polycarbonate and K2Ph [21] or polysulfone and ZIF8 [39]. Mention must be made of the fact that there is no clear evidence in the literature of the relationship between Tg values and chains confinement in the pores. In PPO membranes it was difficult to observe the Tg in the first heating, and for this reason the second heating was recorded. When PPO-based membranes are heated, phase separation occurs at a temperature close to the Tg of the material, as has been reported in the literature for neat polymer [40].

The values supplied by commercial companies and the bulk density of the neat membranes are practically coincident (Table 2). The density of pure PPO was 1.07 g/cm^3^, as has been reported for amorphous PPO [41], which confirmed that the preparation procedure of this membrane yielded an amorphous PPO since crystalline PPO has a higher density of 1.31 g/cm^3^ according to the literature [34]. As was expected, the presence of K2Ph increases the density of the corresponding MMMs, but it can be observed that the experimental bulk density is greater than the theoretical one. This result can be attributed to the fact that part of the polymer chains would remain within the pores of the filler, and that for the same pore volume there is greater mass. 

If we assume that the pores of the filler are totally filled by polymeric chains, the fraction of polymer in the mixed membranes, *ω_P_*, not located in the pores of the filler, can be estimated from the densities of the bare polymer (*ρ_P_*), the mixed matrix membranes (*ρ_MMM_*) and the filler (*ρ_f_*), assuming that the volumes are additives, by the expression [39]:(10)$$\omega_{P}=\frac{\rho_{P}}{\rho_{MMM}}\;\frac{\rho_{f}-\omega_{f}\;\rho_{MMM}}{\rho_{f}}$$

So, the fraction of polymer not occupying the pores of the filler is 0.79 and 0.57 for M@K2Ph with 10% and 20% of load, respectively; 0.59 for PSF@K2Ph-20%; and 0.86 for PPO@K2Ph-5%. This means that nearly all polymer chains are located out of the pores of K2Ph for MMMs with 5% and 10% of filler, whereas almost half of the polymer chains seem to be occupying the pores of the filler in MMMs with 20% of K2Ph. 

### 3.3. Gas Separation Performance

Values of permeability, diffusion and apparent solubility coefficients for oxygen, nitrogen, carbon dioxide, hydrogen, methane and ethylene were determined in all mixed matrix membranes prepared in this work, at 30 °C and 1 bar of pressure. Ethylene was tested for comparative purposes with previous data [21], taking into account the importance of finding highly selective adsorbents not only for CO_2_/C_2_H_4_ separation but also in petrochemical industries.

In a first approximation, Matrimid-based membranes were prepared with 10 and 20% of K2Ph, M@K2Ph-10% and M@K2Ph-20%, respectively, in order to determine the effect of the amount of that filler in the transport properties of appropriately mixed matrix membranes. The results are shown in Table 3. As can be seen, the M@K2Ph-20% membrane showed the highest permeability results for all gasses, so a percentage of 20% of K2Ph was initially selected for PSF and PPO matrices. However, as was previously explained, PPO only admitted a load of 5%. Higher percentages led to a substantial loss of mechanical properties, and the membranes were broken easily.

As can be seen in Table 3, the values of the permeability follow the general trend of P(H_2_) > P(CO_2_) > P(O_2_) > P(C_2_H_4_) >P(N_2_) ≥ P(CH_4_) for all mixed matrix membranes, regardless of what pristine polymer nature or percentage of filler they are. Unfortunately, hardly any permeability data have been reported for other MMMs based on KAP-type fillers. The only data reported are the CO_2_ permeability coefficients of amine-appended KAPs as fillers of Matrimid or polysulphone [20]. The increases in P(CO_2_) of these MMMs, compared with the neat matrices, were higher than those reported in our work, attributed to the chemical interactions between amino moieties and CO_2_ molecules.

To evaluate the flux of gas through a dense membrane it is necessary to take into consideration two contributions: a kinetic parameter (diffusion coefficient, D) and a thermodynamic parameter (solubility coefficient, S).

The diffusive step in glassy polymers could be carried out through channels formed by chain fluctuations, by jumping the diffusant from a cavity or hole to a neighboring one. For this random walk process to take place, the radius of the channels should be at least equal to the Van der Waals radius of the diffusant, σ. Henceforth, one would expect that the activation energy associated with the diffusion process should be proportional to σ^2^, more specifically D ≈ exp(σ^2^) [42]. This means that the plot of Ln D vs. σ^2^ should be a straight line of a negative slope. The kinetic diameter is widely accepted as the size of the diffusant in glassy polymers [43]. In Figure 9 the values of the natural logarithm of the diffusion coefficients of gasses tested through all mixed matrix membranes are plotted as a function of the square of the kinetic diameter. All data fit very well with a straight line, except for CO_2_, probably due to its polar nature, which delays its diffusion through the membranes by the effect of repulsive interactions between the polar polymeric matrices and this gas [44,45], or to the linear configuration of CO_2_, which makes it difficult to represent it as a sphere. Therefore, as can be seen in Table 3 and Figure 9**,** whatever the mixed matrix membrane considered was, the diffusion coefficient follows the trend of D(H_2_) > D(O_2_) > D(CO_2_) > D(N_2_) > D(CH_4_) > D(C_2_H_4_). However, the y-intercept value of each line in Figure 9 varies from one MMM to other. It is clearly seen that the straight line for PSF@K2Ph-20% lies below that of the pristine PSF, presumably as a result of the fact that the addition of 20% of filler to the neat polymer provokes an increase in cohesive energy density, which renders it more difficult to form channels through which the diffusant molecule can slide to a nearby cavity in the MMM than in neat PSF. In other words, the affinity between both phases in this MMM promotes a decrease in gas diffusion by increasing the pathway of the gas through the membrane. However, a lower amount of load did not have any influence on the diffusive steps, as is easily seen when comparing PPO and PPO@K2Ph-5%, or M and M@K2Ph-10%. The addition of 20% of K2Ph to Matrimid lead to a little increase in D, which could be attributed to the fact that the filler enlarges the distance between bulky groups of the Matrimid polymer chains or to the formation of interface voids around the filler, which should be small enough to slightly enhance the free fraction volume. The latter was in agreement with the fact that no significant voids have been detected in SEM micrographs.

The above results are in agreement with a solution–diffusion mechanism. However, to verify this statement the experimental data of all mixed matrix membranes described in this work were fitted to Equation (4). As an example, Figure 10 shows the evolution of the pressure of ethylene in the downstream chamber for the M@K2Ph-20% membrane. The data fitted very well with the isotherm that describes the time dependence of the pressure by the expression resulting from the integration of Fick’s second law. Values of P and D estimated were very similar to those obtained from Equations (6) and (5), respectively, as mentioned previously.

It is generally known that solubility coefficients of gasses in glassy membranes obey the dual-mode model and depend mainly on the gas condensability and the affinity between the filler and polymer, among other factors [46]. Thus, the most (carbon dioxide) and least (hydrogen) condensable gases will be higher and lower solubility parameters, respectively. Indeed, S(CO_2_) > S(C_2_H_4_) > S(CH_4_) > S(O_2_) > S(N_2_) > S(H_2_). As can be seen in Table 3, the values of S rely on the membrane nature in such a way that, in general, S (M-based membranes) > S (PPO-based membranes) > S (PSF-based membranes). In all cases, the addition of K2Ph increases the S coefficients, suggesting that this filler improves the interaction with the gas. This fact was also observed in previous work using polycarbonate as a polymer matrix [21]. However, this interaction varies depending on the percentage of filler added and on the nature of the pristine polymer. It is to say, the addition of 20% of K2Ph to PSF increased the apparent solubility coefficients of bare polymer by up to 380%, while it only increased by up to 130% for Matrimid-based membranes. The addition of 5% of filler to PPO had no effect, while the values of S increased from 40% to 60% when 10% of K2Ph is added to Matrimid. This significant behavior will be examined in future works, which will allow us to take a look into the nature of these interactions, particularly since gas adsorption in Langmuir sites may be associated with the polymer–filler interface, the polymer inside the filler pores and the free volume of the filler not containing polymer.

According to Equation (7), P = D × S, so the variation in the permeability coefficients of all gasses through a dense membrane or hybrid materials can be explained as a balance between gas diffusion and solubility coefficients [47,48]. In the present case, the addition of K2Ph to Matrimid provoked an important increase in the permeability of all gases (Figure 11), mainly attributed to the thermodynamic parameters, as can be seen in Table 3 and Figure 12. The same conclusion applies for the PSF@K2Ph-20% membrane, whose permeabilities rise slightly (see Table 3 and Figure 11) as a consequence of the increase in solubility coefficients, despite the decrease in D compared with the bare polymer (Figure 12). Finally, a very small effect can be observed in PPO mixed matrix membranes (Table 3 and Figure 11).

The performance of a membrane in gas separation can be estimated from the permselectivity coefficient or ideal separation factor, α, a measure of the capacity of a polymeric membrane to carry out the separation of a given gas pair mixture. Values of selectivities of all hybrid membranes described in this work, for different pairs of gasses, are shown in Table 4, together with the neat polymers for comparative reasons.

As can be seen, in the case of Matrimid-based membranes, the addition of K2Ph did not improve the selectivity in gas separations, although the permeability increased significantly in M@K2Ph-20% in comparison with the bare polymer, from 100% for hydrogen to 200% in the case of nitrogen. In PPO-based membranes, there was no variation in permeability nor in permselectivity with the addition of the filler, probably due to the small percentage that this polymer admitted without the loss of its mechanical properties. However, PSF loaded with the K2Ph filler showed similar selectivities to pristine polymer in the separation of certain pairs of gasses, such as O_2_/N_2_ or H_2_/CO_2_, with an increase of 50% in gas permeabilities. It is important to highlight the increase in α (H_2_/CH_4_) and α (H_2_/C_2_H_4_), making this mixed matrix membrane an excellent choice for recovering and purifying hydrogen from refinery and petrochemical industries.

## 4. Conclusions

Twenty percent biphenyl-based knitting aryl polymer (K2Ph) filler is able to increase the permeability coefficients to different gases between two and three times when the matrix is Matrimid (M) and around one-and-a-half times when the matrix is polysulfone (PFS). Besides, this MMM has the highest H_2_/CH_4_ and H_2_/C_2_H_4_ selectivities, making it a good candidate for recovering and purifying hydrogen from petrochemical industries. However, the polyphenylene oxide (PPO) matrix only allows 5% of this load, which does not imply any appreciable effect on the gas transport properties, probably due to the low filler content in this membrane.

## Figures and Tables

**Figure 1 membranes-11-00914-f001:**
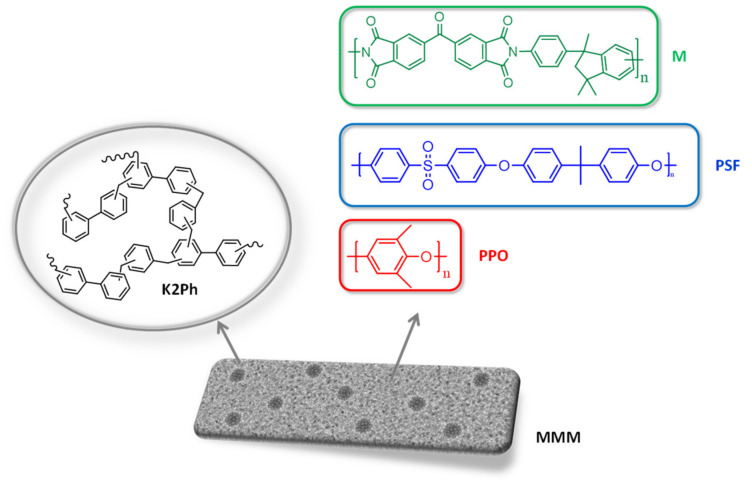
Structures of the biphenyl-based knitting aryl polymer filler (K2Ph) and the matrices used to prepare mixed matrix membranes (MMM): Matrimid (M), polysulphone (PSF) and polyphenylene oxide (PPO).

**Figure 2 membranes-11-00914-f002:**
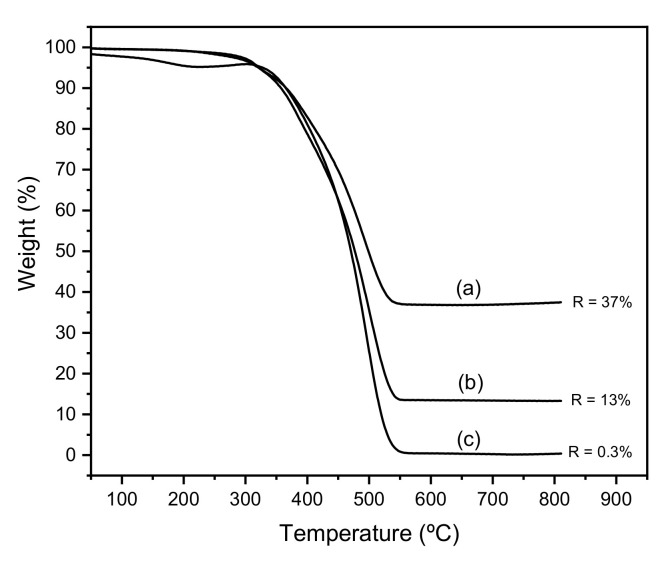
TGAs of K2Ph filler prepared by the Friedel–Craft reaction (**a**) before purification; (**b**) after the first purification cycle; and (**c**) after three purification cycles.

**Figure 3 membranes-11-00914-f003:**
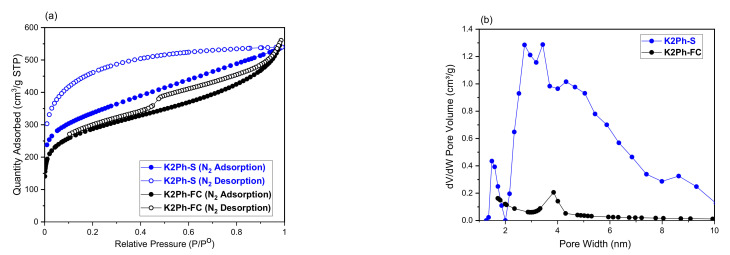
(**a**) N2 adsorption/desorption isotherms and (**b**) the pore size distributions of the K2Ph fillers prepared by the Scheme 2. Ph-S) and the Friedel–Craft reaction (K2Ph-FC, this work).

**Figure 4 membranes-11-00914-f004:**
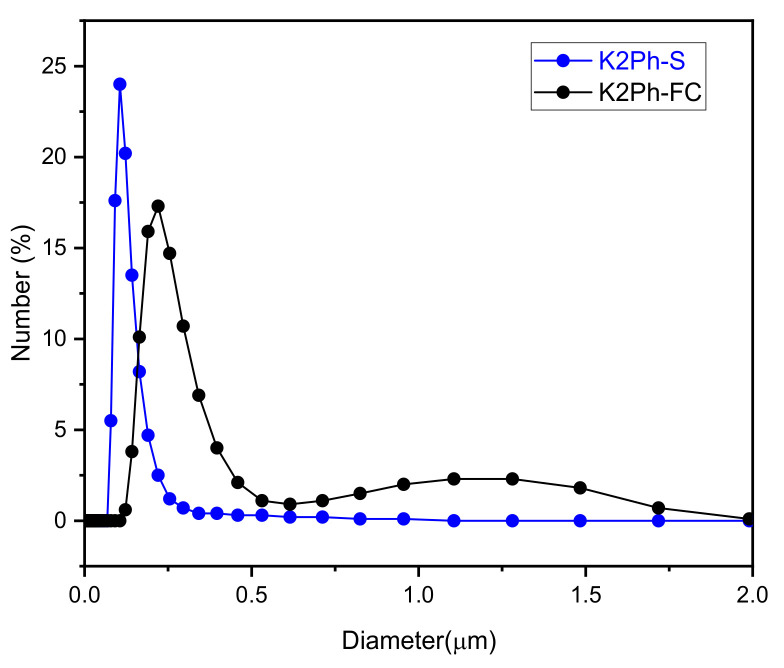
Number average size distribution for K2Ph-S and K2Ph-FC fillers, measured by DLS.

**Figure 5 membranes-11-00914-f005:**
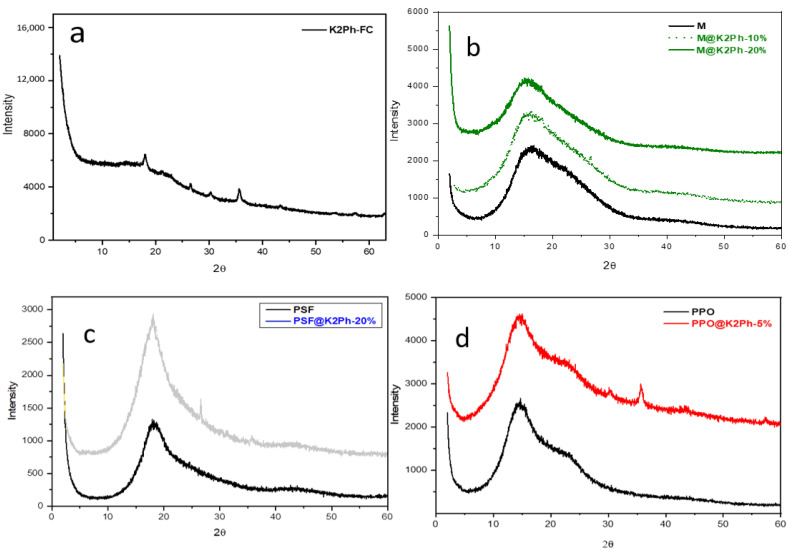
X-ray diffractograms of (**a**) K2Ph filler; (**b**) pristine Matrimid (M) and M-based MMM; (**c**) pristine polysulfone (PSF) and PSF-based MMM; and (**d**) pristine polyphenylene oxide (PPO) and PPO-based MMM.

**Figure 6 membranes-11-00914-f006:**
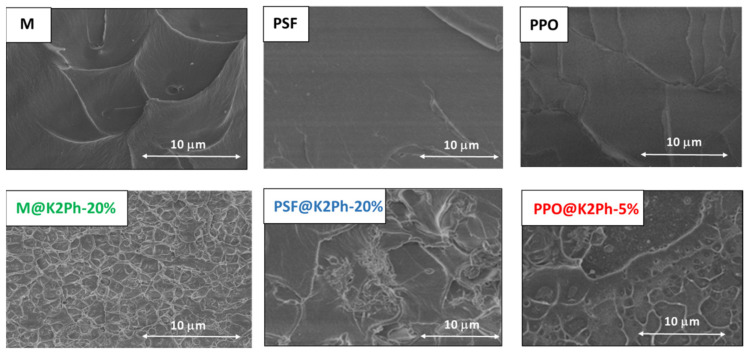
SEM images of the cross-section of pure matrices (**up**) and the corresponding MMMs (**down**).

**Figure 7 membranes-11-00914-f007:**
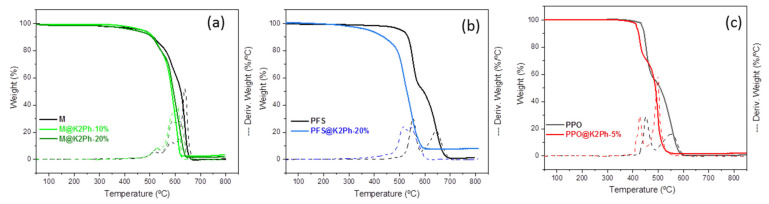
Thermograms of (**a**) pristine Matrimid (M) and M-based MMM; (**b**) pristine polysulfone (PFS) and PFS-based MMM; and (**c**) pristine polyphenylene oxide (PPO) and PPO-based MMM.

**Figure 8 membranes-11-00914-f008:**
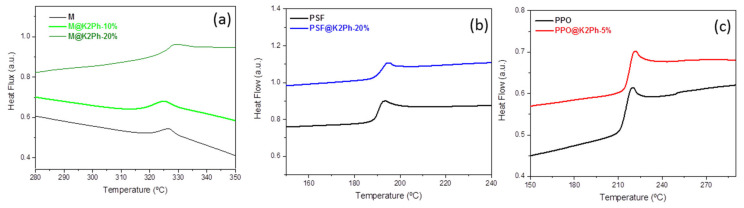
DSC curves of (**a**) pristine Matrimid (M) and M-based MMMs; (**b**) pristine polysulfone (PSF) and PSF-based MMM; and (**c**) pristine polyphenylene oxide (PPO) and PPO-based MMM.

**Figure 9 membranes-11-00914-f009:**
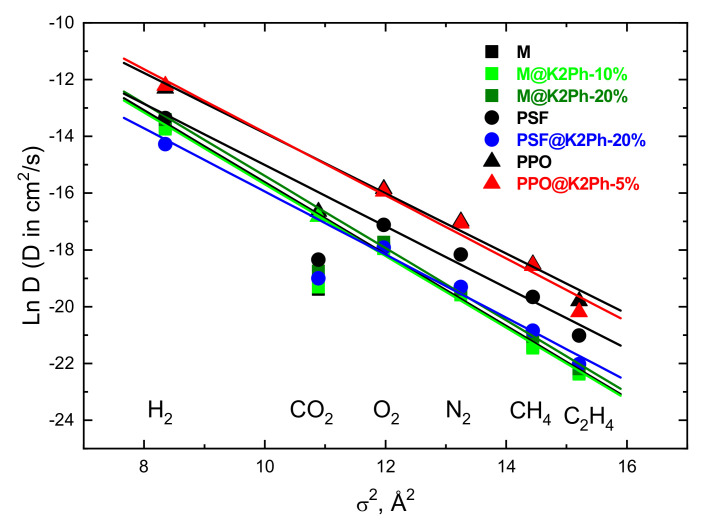
Dependence of the natural logarithm of the diffusion coefficient (Ln D) on the square of the kinetic diameter (σ^2^) of all gasses tested for (squares) Matrimid-, (circles) polysulfone- and (triangles) polyphenylene-oxide-based membranes. The color codes are the following: black symbols for bare polymers, green symbols for MMMs with 5% of filler, red symbols for MMMs with 10% of filler and blue symbols for hybrid membranes with 20% of filler.

**Figure 10 membranes-11-00914-f010:**
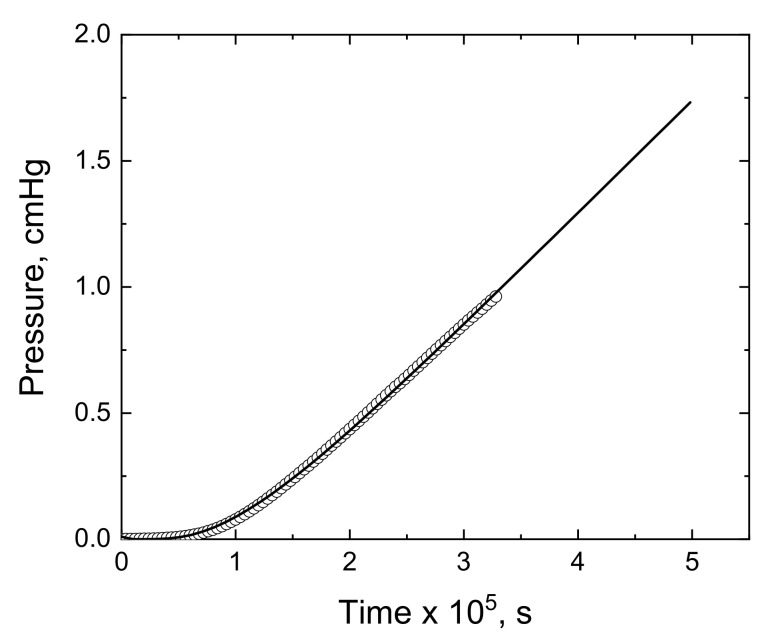
Isotherm showing the variation in the pressure of ethylene in the downstream chamber with time for the M@K2Ph-20% membrane. The continuous line represents the time dependence as calculated from the integration of Fick’s second law represented in Equation (4).

**Figure 11 membranes-11-00914-f011:**
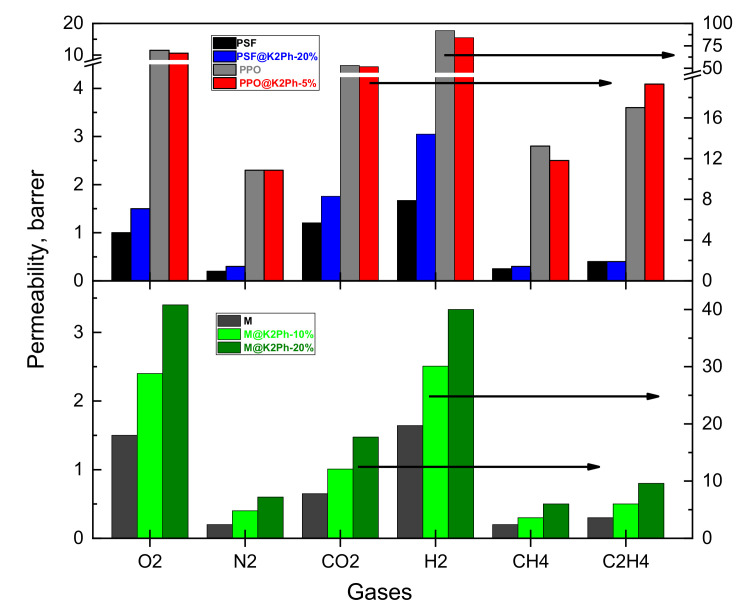
Graphic representation of permeability coefficients at 30 °C and 1 bar of pressure of MMMs of this work and the corresponding pristine membranes.

**Figure 12 membranes-11-00914-f012:**
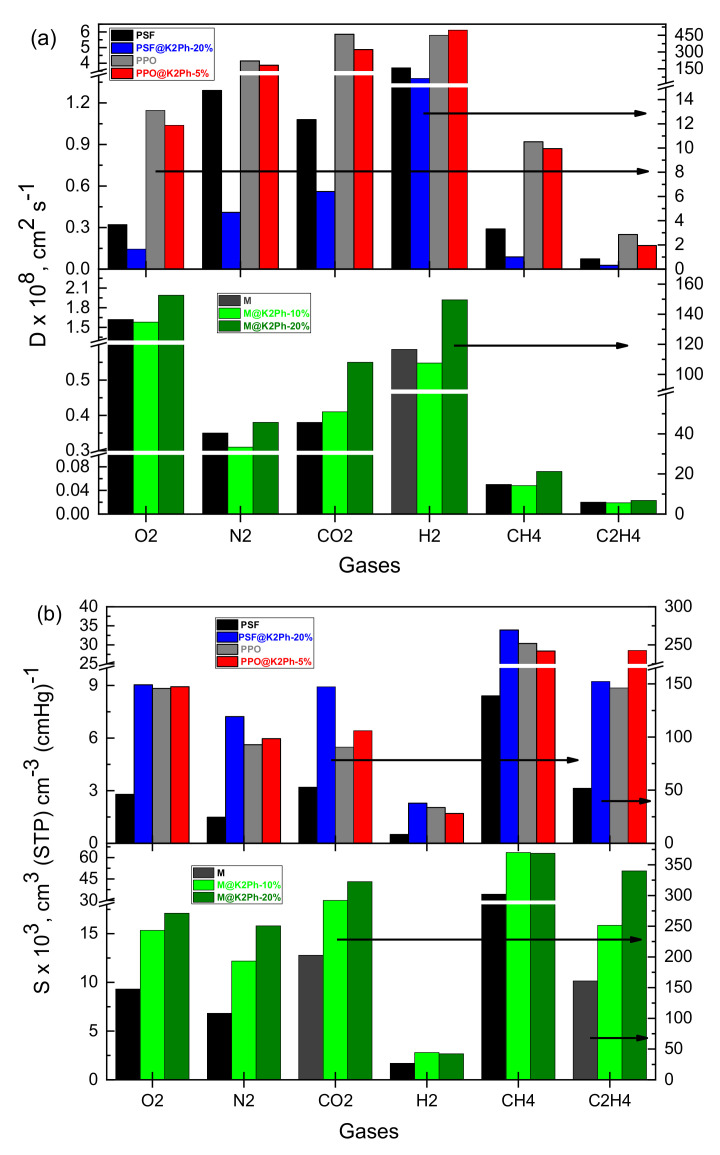
Graphic representation of (**a**) diffusion and (**b**) solubility coefficients at 30 °C and 1 bar of pressure of MMMs of this work and the corresponding pristine membranes.

**Table 1 membranes-11-00914-t001:** Textural properties and density of K2Ph polymeric fillers.

Filler	S_BET_ (m^2^ g^−1^)	Pore V_total_ (cm^3^ g^−1^) ^a^	ΔPore Size (nm)	Skeletal Density (g cm^−3^)	Bulk Density (g cm^−3^)	V_op_(%) ^b^
K2Ph-S	1481	0.904	2.44	1.246 ^c^	0.586 ^c^	53 ^c^
K2Ph-FC	1034	0.867	3.35	1.312	0.614	53

^a^ P/P_0_ = 0.98, 77°K ^b^ Volume occupied by the pores in the filler. ^c^ Data taken from [26].

**Table 2 membranes-11-00914-t002:** Thermal properties and densities of pristine polymers and mixed matrix membranes prepared with polysulfone (PSF), Matrimid (M) and polyphenylene oxide (PPO), with different percentages of K2Ph filler.

Membrane	Td (°C)	Tg	Skeletal Density ^a^	Bulk Density ^a^	
				Experimental	Calculated
M	640,584	326	-	1.194	1.20 ^c^
M@K2Ph-10%	615,588	330	1.100	1.214	1.095
M@K2Ph-20%	615,588	327	1.036	1.251	1.008
PSF	530	190	-	1.231	1.24 ^b^
PSF@K2Ph-20%	517,546	193	1.013	1.242	0.98
PPO	452,555	216	-	1.070	1.06 ^d^
PPO@K2Ph-5%	427,500	218	0.925	1.124	1.023

^a^ In g cm^−3^; ^b, c, d^ values supplied by Sigma-Aldrich, Ciba-Geigy and Poly Sciences, respectively.

**Table 3 membranes-11-00914-t003:** Values of the permeability, diffusion and apparent solubility coefficients of different gasses, at 30 °C and 1 bar of pressure for all mixed matrix membranes prepared in this work. Data for pristine polymers (M, PSF and PPO) are also included.

Membrane (Thickness, µm)	Gas	P, Barrer ^a^	D × 10^8 b^	S × 10^3 c^
M (99 ± 1)	O_2_	1.5	1.6	9.3
	N_2_	0.2	0.3	6.8
	CO_2_	7.8	0.4	202.8
	H_2_	19.7	116.6	1.7
	CH_4_	0.2	0.05	34.3
	C_2_H_4_	0.3	0.02	160.9
M@K2Ph-10% (80 ± 1)	O_2_	2.4	1.6	15.3
	N_2_	0.4	0.3	12.2
	CO_2_	12.1	0.4	292.0
	H_2_	30.1	107.5	2.8
	CH_4_	0.3	0.05	63.5
	C_2_H_4_	0.5	0.02	251.3
M@K2Ph-20% (124 ± 7)	O_2_	3.4	2.0	17.1
	N_2_	0.6	0.4	15.8
	CO_2_	17.7	0.6	322.5
	H_2_	40.0	149.5	2.7
	CH_4_	0.5	0.07	63.0
	C_2_H_4_	0.8	0.02	339.9
PSF (64 ± 2)	O_2_	1.0	3.7	2.8
	N_2_	0.2	1.3	1.5
	CO_2_	5.7	1.1	52.9
	H_2_	7.9	158.1	0.5
	CH_4_	0.25	0.3	8.4
	C_2_H_4_	0.4	0.07	51.9
PSF@K2Ph-20% (70 ± 2)	O_2_	1.5	1.6	9.0
	N_2_	0.3	0.4	7.2
	CO_2_	8.3	0.6	147.3
	H_2_	14.4	62.9	2.3
	CH_4_	0.3	0.09	33.9
	C_2_H_4_	0.4	0.03	152.3
PPO (128 ± 2)	O_2_	11.5	13.1	8.8
	N_2_	2.3	4.1	5.6
	CO_2_	53.0	5.9	90.5
	H_2_	91.9	449.9	2.0
	CH_4_	2.8	0.9	30.3
	C_2_H_4_	3.6	0.3	146.1
PPO@K2Ph-5% (129 ± 2)	O_2_	10.6	11.9	8.9
	N_2_	2.3	3.9	6.0
	CO_2_	51.6	4.9	105.9
	H_2_	84.0	495.5	1.7
	CH_4_	2.5	0.9	28.4
	C_2_H_4_	4.09	0.2	242.6

^a^ Barrer = 10^−10^ cm^3^ (STP) cm cm^−2^ s^−1^ (cmHg)^−1^; ^b^ D in cm^2^ s^−1^; and ^c^ S in cm^3^ (STP) cm^−3^ (cmHg)^−1^.

**Table 4 membranes-11-00914-t004:** Permselectivity coefficients (α) for different pairs of gases in all the mixed matrix membranes and in the pristine polymers, at 30 °C.

Membrane	O_2_/N_2_	CO_2_/N_2_	CO_2_/CH_4_	CO_2_/C_2_H_4_	H_2_/CO_2_	H_2_/CH_4_	H_2_/C_2_H_4_	C_2_H_4_/CH_4_
M	7.5	39.0	39.0	26.0	2.5	98.5	65.7	1.5
M@K2Ph-10%	6.0	30.2	40.3	24.2	2.5	100.3	60.2	1.7
M@K2Ph-20%	5.7	29.5	35.4	22.1	2.3	80.0	50.0	1.6
PSF	5.0	28.5	22.8	14.3	1.4	31.6	19.8	1.6
PSF@K2Ph-20%	5.0	27.7	27.7	20.8	1.7	48.0	36.0	1.3
PPO	5.0	23.0	18.9	14.7	1.7	32.8	25.5	1.3
PPO@K2Ph-5%	4.6	22.4	20.6	12.6	1.6	33.6	20.5	1.6

## Data Availability

All available data is collected in the article.

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
