# Peer review of "Mixed Matrix Membranes Containing a Biphenyl-Based Knitting Aryl Polymer and Gas Separation Performance"

_membranes, 2021, doi:10.3390/membranes11120914_

Round 1

Reviewer 1 Report

  1. In section 3.1, what are the sizes of the K2Ph fillers?
  2. How does the viscosity of the polymer solution was affected by adding the filler? Please provide the data for viscosity.
  3. Does K2Ph filler interacts with the polymer matrix?
  4. Please label the d-spacing in figure 5. Also, please discuss the d-spacing of the K2Ph filler in the manuscript.
  5. The distribution of the filler in the matrix on figure 6 is suggested to identify using EDX to map the presence of the particle,
  6. How many times does the author measures the permeability of the membranes? Standard deviation is suggested to include.
  7. Please compare your membrane performance with other related literature.

Author Response

Point-by-point responses are detailed in the attached document

Reviewer 2 Report

The paper is generally well written and presented. I have only a few comments:

  • It is not clear how D is calculated from equation 4, the previous text describes p vs t but in the expression of p(t) I do not see any theta parameter?
  • Values of diffusion coefficients seems rather small, what would the typical literature values be for similar systems? This is not clearly highlighted in the text. It discusses about ethylene comparison with previous data but does not give numbers. The pore size is not that small hence I would expect higher diffusion coefficients.
  • I would also expect to be in Knudsen diffusion regime but the authors seem not to discuss this point? The Knudsen diffusion equation could perhaps be used to obtain predictions of diffusion and see how they compare with the values calculated with Equation 4.
  • The authors are missing some relevant literature, including a very recent work of Tanaka et al., Chemical Engineering Journal 424 (2021) 129313

The paper can be considered for publications once the above points have been considered.

Author Response

The point-by-point responses are detailed in the attached document. The English language and style have been improved. 

Reviewer 3 Report

This paper discuss gas separation performance of the MMMs using K2Ph POP as additive and Matrimid, PSF, PPO as matrices. The work is quite impressive, and the study is very systematically. Couple quick questions before it can be published. 

  1. Just for my curious, can author make a comments on the pattern showing on the SEM images in your MMMs.
  2. Can author provide the sorption and TGA data for the pure K2Ph? 
  3. It is little surprising to me that the CO2 diffusivity is such low, even lower than N2? 

Author Response

(The authors gave the same response as above.)

Round 2

Reviewer 1 Report

The authors addressed my concerns well.